# Combating Visual Question Answering Hallucinations via Robust Multi-Space Co-Debias Learning

Jiawei Zhu*
Beijing Institute of
Technology, Zhuhai
Zhuhai, China
zhujiawei@m.scnu.edu.cn

Yishu Liu*
Harbin Institute of
Technology, Shenzhen
Shenzhen, China
liuyishu@stu.hit.edu.cn

Huanjia Zhu
South China Normal
University
Guangzhou, China
zhuhuanjia@m.scnu.edu.cn

Hui Lin†
China Academic of
Electronics and
Information Technology
Beijing, China
linhui@cetc.com.cn

Yuncheng Jiang
South China Normal
University
Guangzhou, China
ycjiang@scnu.edu.cn

Zheng Zhang
Harbin Institute of
Technology, Shenzhen
Shenzhen, China
darrenzz219@gmail.com

Bingzhi Chen†
Beijing Institute of
Technology, Zhuhai
Zhuhai, China
chenbingzhi.smile@gmail.com

## ABSTRACT

The challenge of bias in visual question answering (VQA) has gained considerable attention in contemporary research. Various intricate bias dependencies, such as modalities and data imbalances, can cause semantic ambiguities to generate shifts in the feature space of VQA instances. This phenomenon is referred to as **"VQA Hallucinations"**. Such distortions can cause hallucination distributions that deviate significantly from the true data, resulting in the model producing factually incorrect predictions. To address this challenge, we propose a robust Multi-Space Co-debias Learning (MSCD) approach for combating VQA hallucinations, which effectively mitigates bias-induced instance and distribution shifts in multi-space under a unified paradigm. Specifically, we design bias-aware and prior-aware debias constraints by utilizing the angle and angle margin of the spherical space to construct bias-prior-instance constraints, thereby refining the manifold representation of instance de-bias and distribution de-dependence. Moreover, we leverage the inherent overfitting characteristics of Euclidean space to introduce bias components from biased examples and modal counterexample injection, further assisting in multi-space robust learning. By integrating homeomorphic instances in different spaces, MSCD could enhance the comprehension of structural relationships between semantics and answer classes, yielding robust representations that are not solely reliant on training priors. In this way, our co-debias paradigm generates more robust representations that effectively mitigate biases to combat hallucinations. Extensive experiments

on multiple benchmark datasets consistently demonstrate that the proposed MSCD method outperforms state-of-the-art baselines.

## CCS CONCEPTS

• **Computing methodologies** → *Natural language generation*; *Spatial and physical reasoning*.

## KEYWORDS

Visual Question Answering, VQA Hallucinations, Robust Learning, Multi-Space Learning

**ACM Reference Format:**
Jiawei Zhu, Yishu Liu, Huanjia Zhu, Hui Lin, Yuncheng Jiang, Zheng Zhang, and Bingzhi Chen. 2024. Combating Visual Question Answering Hallucinations via Robust Multi-Space Co-Debias Learning. In *Proceedings of the 32nd ACM International Conference on Multimedia (MM '24), October 28-November 1, 2024, Melbourne, VIC, AustraliaProceedings of the 32nd ACM International Conference on Multimedia (MM'24), October 28-November 1, 2024, Melbourne, Australia.* ACM, New York, NY, USA, 10 pages. https://doi.org/10.1145/3664647.3681663

## 1 INTRODUCTION

The visual question answering (VQA) task aims to build an agent proficient in collaborative reasoning, leveraging question reasoning and image semantics for making informed predictions. Previous methods have shown significant performances [1, 2, 37], which typically train VQA models on training data and test on benchmark datasets with similar distributions. Regrettably, contemporary research reveals a significant challenge: many methods experience a notable decrease in model generalization performance when the test data deviates from the training distribution. Indeed, some studies attempt to address biases in modality (language, vision) and data distribution. Through the exploration of these biases [5, 11, 18–21, 30], it becomes evident that biases in distribution cause the model to learn idiosyncratic biases closely linked to specific modalities and data labels, rather than focusing on the holistic semantics of the vision-question instance. This phenomenon is termed **"VQA hallucination"** problem, as shown in Figure 1. It can be observed that

---

*Both authors contributed equally to this research.

†Corresponding authors: Bingzhi Chen and Hui Lin.

---

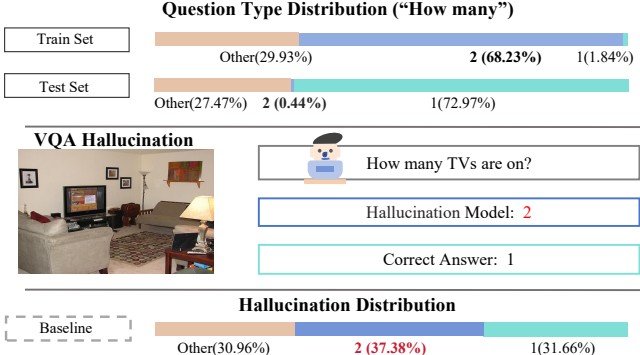

**Figure 1: Illustration of the VQA Hallucinations: given a question type ("How many..."), we display the answer distributions for the ground-truth answers from the train and test set. The model suffers from factual hallucinations [17, 27]. An intriguing example of this phenomenon is the emergence of an improbable "2" distribution within VQA-CP v2 dataset [1].**

these biases can cause instances and distribution shifts [24, 33, 51] in the feature space to produce hallucination distributions.

To further investigate the relationship among biases, instance and distribution shifts, and VQA hallucination, we visualize the feature space divided by question types in Figure 2. In conventional learning methods, semantic ambiguities in instance and distribution shifts become apparent. For instance, considering the question type "What time…": certain features that should predict answers like "morning" or "night" (which have fewer occurrences in the training) are incorrectly clustered with "afternoon" in the feature space. These shifts negatively affect the out-of-distribution (*OOD*) robustness performance of the model.

In the literature, mainstream VQA methods mainly focus on language priors and can currently be categorized into three types: ensemble model-based, data augmentation-based models, and feature space-based models. These approaches encompass the following components: introducing an ensemble bias model [5, 10, 11, 20, 21] to identify and mitigate biases inherent in each modality or dataset, employing manual annotations or data augmentation techniques [7, 8, 31] to mitigate language priors, and utilizing models that focus on fine-grained feature space learning [4, 18, 22] to address biases based on the frequency or distribution of instances in the feature space. While these methods have shown promise in bolstering robustness, their approach of solely targeting partial biases without considering intrinsic patterns has led to suboptimal performance.

This work rethinks the relationship between biases, instance and distribution shifts, and hallucination, and attempts to understand "Why do traditional visual question answering methods produce hallucination problems on *OOD* data". Given the characteristics of feature space, methods relying on Cross-Entropy (CE) loss face challenges in accurately delineating decision boundaries [4, 13, 18, 36, 52] within the feature space, where the span of features for each class needs to be proportional to the corresponding number of instances. In scenarios where the distributions of train and test sets are similar, the model can effectively discern the separability of each instance. However, training in a biased environment leads the

model to prioritize fitting the dominant majority answer class while overlooking the minority answer class outside the distribution. This runs counter to the original intention of visual question answering to understand vision-question clues and reason about answers based on semantics. Consequently, instances from the minority class in the test set may experience a shift, leading to a distribution shift. In such cases, the model fails to attend to the semantics of the instance itself and resorts to making arbitrary predictions. This work explains the hallucination phenomenon from the instance and distribution shift caused by biases.

Building upon this conceptual framework, our objective is to mitigate the phenomenon of hallucinations, stemming from modal shortcuts and imbalanced dataset distributions. Leveraging the intrinsic patterns and shared sample structures, we aim to unlock the optimal de-hallucination potential of vision-question instances through a multi-space homeomorphism perspective. Simultaneously, we intend to construct bias examples and modal counterexamples to further refine bias learning and counteract biases more effectively. As illustrated in Figure 3, our proposed Multi-Space Co-Debias (MSCD) method initiates from a Spherical space comprised of angles, employing bias-aware and prior-aware debias constraints. This approach explicitly calibrates instances to generate discriminative manifold representations, while alleviating prior distribution dependence and improving robustness to combat hallucinations caused by instance and distribution shifts. Furthermore, we harness the fitting characteristics of Euclidean space (utilizing softmax CE loss) to introduce biases components from bias examples and modal-counterexamples, aiding the model in robust learning. The primary contributions of our work can be summarized as follows:

- This paper introduces a novel multi-space co-debias learning to address the issue of VQA hallucination stemming from biases. It tackles the problem by Spherical and Euclidean spaces co-debias learning into a unified framework.
- Two novel bias-aware and prior-aware debias constraints are designed for spherical debias learning and explicitly construct constraints to calibrate instance shift and distribution shift, thereby alleviating VQA hallucinations.
- A well-designed multi-space co-debias paradigm is proposed by deploying a two-stage strategy of Euclidean space, assisting spherical debias learning to expose prior correlations and modality-semantics interplay.
- Extensive experiments on two biased benchmark datasets and a balanced dataset demonstrate the effectiveness of the proposed MSCD method to combat VQA hallucination and achieve state-of-the-art performance.

## 2 RELATED WORK

### 2.1 Robustness Methods of VQA Tasks

Although performance in the VQA task has approached human-level performance, susceptibility to VQA hallucinations persists, resulting in insufficient robustness. The introduction of the new benchmark dataset VQA v2 [16], complementing VQA v1 [3], includes question-image pairs with similar semantics and diverse answers. The emergence of bias datasets such as VQA-CP v1, VQA-CP v2 [1] , which use different protocols for VQA datasets, provides benchmarks for debias methods. Notably, the answers to the *train*

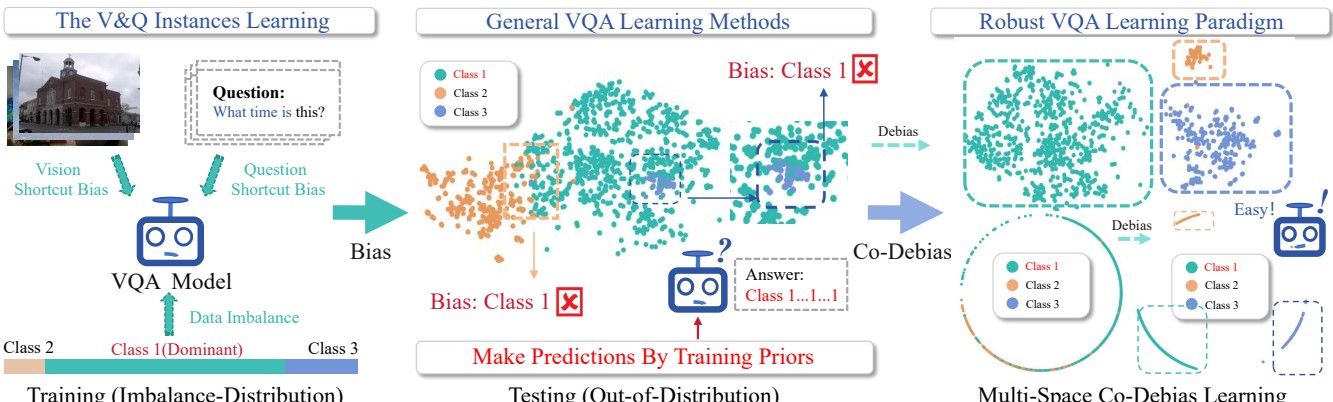

**Figure 2: Illustration of robust VQA learning paradigm. The general VQA learning process fits the prior in training, which cannot distinguish mixed bias instances well, compromising the robustness of the model out-of-distribution. Under the multi-space robust learning paradigm, the model can better bridge the instance-semantic feature space for real reasoning.**

set and *test* set are inversely distributed, making it convincing to evaluate the model against the hallucination problem. Previous research predominantly addresses shortcut biases originating from modalities (question or image) and imbalance biases within datasets. Existing research on robustness primarily focuses on various methods to mitigate bias, including addressing modality shortcut bias [19, 34, 38, 45], dataset bias [7, 28, 31, 50], and comprehensive bias mitigation strategies [4, 18, 20, 21, 25]. We summarize and discuss these works as follows:

**Methods that addressing modality interplay:** Collaborative reasoning between visual and textual cues is pivotal for fostering robust learning in visual question answering [19, 20, 22, 34, 38, 45]. Recent literature delves into this issue by examining theoretical frameworks such as causal inference [39] and confounding factors [44]. Discuss with the instance, the PW-VQA [45] delves into the confounding effect arising from the interplay of vision and language through a causal lens, shedding light on potential biases in both modalities. Similarly, CVIV [38] employs IVar to amplify visual features, enabling the model to pinpoint essential visual cues. These methods offer valuable insights into the collider bias phenomenon within the realm of vision-language interaction.

**Methods to balance dataset bias:** Existing robust learning frameworks attempt to capture specific dataset biases through annotations. The CSS-based methods, as represented by [7, 28, 31], generates a plethora of counterfactual samples to rectify data bias. This is achieved by strategically masking key objects in images or words in questions. Additionally, certain studies adopt strategies that don't rely on extensive annotation. The D-VQA [50] employs negative samples and branch detection modules to tackle bias both at the feature and sample levels. These methods explore the effectiveness of changing and adapting when distribution priors change.

**Methods that comprehensive bias mitigation:** Given the elusive and intricate nature of bias, the development of comprehensive debias methods has been paramount. These approaches [10, 20–22, 25] have yielded promising results, showcasing their potential to mitigate bias effectively. The GGE [20] and GGD [21] framework employ a greedy training strategy on both the biased

and base models. This approach systematically infuses bias information to aid the model in bias mitigation. The Ensemble-based models represented by GenB [10] alleviate the bias in the basic model by training a bias model.

Different from the above methods, we seek to co-debias constraints on the homeomorphism of multi-spaces to combat hallucinations. Inspired by [4, 18, 22, 43, 47], we solve the inherent pattern of hallucinations from the root by exploring a new multi-feature space paradigm. That is, this poor out-of-distribution performance is caused by instance shift and distribution shift during training. Considering the geometric characteristics of VQA data, we propose a unified perspective that utilizes Euclidean space and Spherical space co-debias to solve the VQA hallucination problem.

### 2.2 Spherical Space Learning

By harnessing the capabilities of Spherical space learning to bolster instance discrimination and robustness, many works have adopted the normalization of embeddings to the unit hypersphere [23, 35, 41, 42, 55, 56]. Spherical space learning involves constructing an angular space through a regularized classification function, thereby establishing more rational decision boundaries. Among these notable recent works, A-Softmax [35] was developed to refine discriminative face embeddings, enhancing feature discriminability by directly connecting with hypersphere manifolds. HSME [23] focuses on identifying visible thermal human bodies by leveraging hypersphere manifolds and employing metric learning to attain distinct and robust feature representations. Recently, inspiring research in visual question answering has emerged. AdaVQA [18] pioneered the application of angular space to tackle the language prior problem via feature space learning, effectively mitigating bias. RMLVQA [4] proposed a methodology utilizing adaptive margin loss to improve robustness to bias, taking into account answer difficulty and frequency. Different from these approaches, this work employs instance and distribution shifts to mitigate correlations among deep representations distributed on the hypersphere. This strategy enhances the instance discriminability and unbinds it from the prior distribution, ultimately alleviating the VQA hallucinations caused by instance shift and distribution shift.

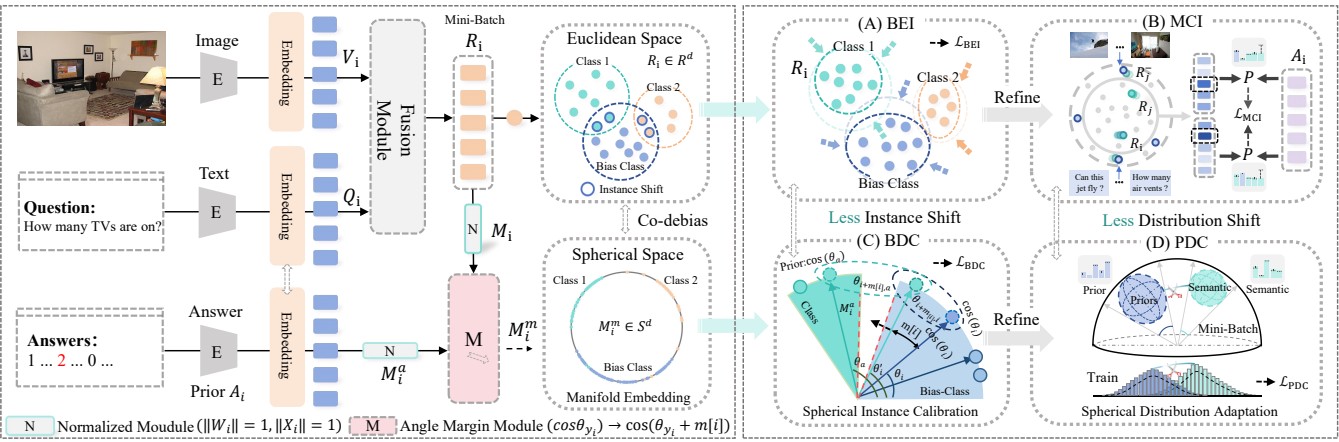

**Figure 3: Our robust VQA framework is built on a multi-space co-debias learning framework for combating VQA hallucinations. A multi-space co-debias learning is built through the instance fusion module and the angle margin module, and the robustness representation is obtained by co-debias the purified Spherical space and Euclidean space.**

## 3 METHODOLOGY

### 3.1 Problem Formulation

Following previous research [1, 4, 18], the VQA task essentially acts as a multi-label classification problem. Without loss of generality, given a batch of data samples, $\mathcal{D} = \{(\mathcal{V}_i, \mathcal{Q}_i), \mathcal{A}_i\}_{i=1}^N$, where $(\mathcal{V}_i, \mathcal{Q}_i)$ is the $i$-th image-question pairs of samples with the corresponding ground-truth answer $\mathcal{A}_i$, and $N$ is the number of samples. We need to optimize the mapping function $\mathcal{F}_{vqa} : \mathcal{V} \times \mathcal{Q} \to \mathcal{R}$ to get a joint representation $\mathcal{R}$. The VQA base model embeds the two features for fusion to obtain a joint representation:

$$\mathcal{R}_i = \mathcal{F}_{vqa}(\mathcal{V}_i, \mathcal{Q}_i; \theta_m) = f_\theta\left(e^v(\mathcal{V}_i), e^q(\mathcal{Q}_i)\right), \quad (1)$$

where $\mathcal{R}_i$ denotes the joint representation of the $i$-th instance, $f_\theta(\cdot)$ is the joint network with parameters $\theta_m$, $e^v(\cdot)$ is pretrained image encoder, and $e^q(\cdot)$ is pretrained question encoder. As such, the objective function is depicted as:

$$\hat{\mathcal{A}} = \arg\max p\left(\mathcal{A}_k \mid \mathcal{R}_i; \theta_c\right), \quad (2)$$

where $\theta_c$ represents the parameters of the answer classifier, and $\mathcal{A}_k$ denotes the $k$-th answer in the answer set $|\mathcal{A}|$. Note that each instance probably has multiple correct answers, hence the optimization objective of training the model can be written as:

$$\mathcal{L}_{VQA}\left(\hat{\mathcal{A}}_i, \mathcal{A}\right) = -\frac{1}{D} \sum_{i=1}^D \sum_{k=1}^{|\mathcal{A}|} \mathcal{S}_{(i,k)} \log\left(p\left(\mathcal{A}_k \mid \mathcal{A}_i\right)\right), \quad (3)$$

where $\mathcal{S}_{(i,k)}$ is the score of the $i$-th instance corresponding to the $k$-th answer of the answer candidates. Through training, our goal is to optimize a network $\mathcal{F}_{vqa}(\theta)$ that maximizes $OOD$ performance:

$$\max_{m, \theta_m, \theta_c} \left(\varepsilon_{OOD}\left(\mathcal{F}_{vqa}(\mathcal{V}_i, \mathcal{Q}_i; \theta_m, \theta_c)\right)\right). \quad (4)$$

Among them, $\varepsilon_{OOD}$ represents robustness evaluation on $OOD$ data.

### 3.2 Multi-Space Representation Mapping

As mentioned before, the MSCD is combined to calibrate instances and distribution, thereby achieving VQA robustness in distribution imbalance. In the following, different spaces have the same sample structure, and we first introduce the loss function for robust multi-space co-debias learning.

**Euclidean Feature Space:** Euclidean space represents the most common zero-curvature manifold [12]. In feature space, the two points can be represented on the plane. The commonly used cross-entropy (CE) loss separates features of different classes by maximizing the posterior probability of the ground truth class, but suffers from insufficient class discriminability [18, 35, 41]. The $E(\cdot)$ model can be trained by optimizing the CE loss as follows:

$$\mathcal{L}_{CE} = \frac{1}{N} \sum_{i=1}^{|\mathcal{A}|} -\log P_i = \frac{1}{N} \sum_{i=1}^{|\mathcal{A}|} -\log \frac{e^{\mathcal{W}_{a_i}^T \mathcal{R}_i + b_i}}{\sum_{j=1}^C e^{\mathcal{W}_j^T \mathcal{R}_i + b_j}}, \quad (5)$$

where $P_i$ denotes the posterior probability of the $i$-th instance representation $R_i$ when classified under label $a_i$. The $b_i$ and $b_j$ refer to the weight biases for answer classes $i$ and $j$. Here, $N$ represents the total number of instances and $C$ signifies the number of classes.

**Spherical Feature Space:** To optimize the instance representation space, many studies [35, 52, 57] have been conducted to enhance intra-class compactness and inter-class separation, making the representation more discriminative and the angular representations located on the unit sphere space a more reliable classification metric. Following the previous setting [4, 18], through instance representation regularization, the answer representations of visual question instances can be converted from Euclidean space to angular space, forming a spherical space. Specifically, we initialize the angle margin by utilizing $L_2$-normalization of the weight vector $\mathcal{W}_i$ and the joint representation $\mathcal{R}_i$ to ensure that the posterior probability is determined by the angle $\theta_i$. Let $\theta_i$ be the angle between $\mathcal{R}_i$ and $\mathcal{W}_i$. Therefore, the logits for each instance are transformed as:

$$f_i = \mathcal{W}_i^\top \mathcal{R}_i = \|\mathcal{W}_i\| \|\mathcal{R}_i\| \cos\theta_i = s(\cos\theta_i), \quad (6)$$

where the instance representation $\mathcal{R}_i$ ($\|W_i\| = 1$, $\|R_i\| = 1$, $b_{i,j} = 0$) is thus distributed on a hypersphere with a radius $s$. This makes the normalized loss as:

$$\mathcal{L}_{SPH} = \frac{1}{N} \sum_{i=1}^{|\mathcal{A}|} -\log\left(\frac{e^{s\cos(\theta_{a_i,i})}}{\sum_{j=1}^C e^{s\cos(\theta_{j,i})}}\right). \quad (7)$$

Although we can learn features with angular boundaries through sphere loss, these representations are still not necessarily discriminative [18, 35, 42].

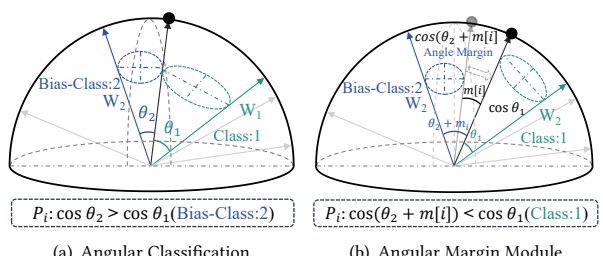

$$\boxed{P_i: \cos\theta_2 > \cos\theta_1 (\text{Bias-Class:2})} \qquad \boxed{P_i: \cos(\theta_2 + m[i]) < \cos\theta_1 (\text{Class:1})}$$

(a) Angular Classification      (b) Angular Margin Module

**Figure 4: Illustration of predictions in Spherical space. We have simplified the categories for readability.**

### 3.3 Spherical Debias Learning

Under the supervision of Spherical learning, instances have geometrically interpretable representations. Different from previous work, we do not modify the loss function [4, 18, 35, 52].

**Angle Margin Module (AMM):** In VQA, for the predictions of any image-question instances, the classification result depends on the angle $\theta$, as shown in Figure 4(a). Giving a motivating VQA binary-class example as a case, regarding the posterior probability $P_i$ (determined by cosine distance) of the predicted feature of the $i$-th instance, the final result only depends on the angles $\theta_1$ and $\theta_2$ and the maximum value of cosine distance $P_i$ can be obtained as the classification result, and it is extended to VQA multi-class task. Therefore, by mapping these representations onto a hypersphere manifold and distinguishing them by angle. Further, following the previous settings [4, 18], an adaptive angle margin $m_i$ is introduced to effectively change the classification decision through the angle margin, as shown in Figure 4(b). It is formalized as follows:

$$L_{\text{SPH}-M} = \sum_{i=1}^{|\mathcal{A}|} -a_i \log \frac{\exp\left(s\cos\left(\theta_i + m[i]\right)\right)}{\sum_{j=1}^{|\mathcal{A}|} \exp\left(s\cos\left(\theta_j + m[j]\right)\right)}, \quad (8)$$

where $m[i]$ is the adaptive instance angle margin. Adding angle margin $(\theta_i + m[i])$ to the manifold representations enables the model to distinguish between frequent/rare instances. However, the feature space changes dynamically. The feature space of frequently occurring instances is narrower than the original feature space. Subtle biases will further the instances shift, causing the feature space of the class space of frequent instances to become more crowded, and the model has hidden dangers of invisible hallucinations. To address this limitation, this work further constructs a Spherical debias learning method to eliminate biases.

**Bias-Aware Debias Constraint (BDC):** In the Spherical space, adding the angle margins module can help the model learn instance-specific to improve instance discriminability. In fact, the existence of biased instances still damages the effectiveness of semantic learning of instances. Inspired by [9, 15, 48], we explicitly construct bias-aware debias constraints, which prompt the learned instances to further alleviate the impact of biased instances and focus on the learning of image-question semantics. Intuitively, The $\mathcal{M}_i = s\cos(\theta_i)$ representation without margins is more likely to contain biased information, as shown in Figure 4(a). Simultaneously, we introduce prior which $\mathcal{R}_i^a = e^a(\mathcal{A}_i)$ from the corresponding answer prior, which is further transformed $\mathcal{M}_i^a$ into angle space:

$$\mathcal{M}_i^a = \left\| \mathcal{W}_i^a \right\| \left\| \mathcal{R}_i^a \right\| \cos\theta_i^a = s\left(\cos\theta_i^a\right). \quad (9)$$

This work further optimizes the learned manifold representation $\mathcal{M}_i^m$. Specifically, the goal of **BDC** is to bring the manifold representation $\mathcal{M}_i^m$ as close as possible to the prior $\mathcal{M}_i^a$ while pushing away from the biased $\mathcal{M}_i$.

Inspired by [14, 53], we consider generalizing the maximization formula of state entropy to construct bias-aware debias constraint form as follows:

$$L_{\text{BDC}} = \log \frac{e^{\kappa\langle z_i, z_a\rangle}}{e^{\kappa\langle z_i, z_a\rangle} + \sum_{k\neq i} e^{\kappa\langle z_i, z_b\rangle}}$$

$$= \kappa\langle z_i, z_a\rangle - \log\left(e^{\kappa\langle z_i, z_a\rangle} + \sum_{k\neq i} e^{\kappa\langle z_i, z_b\rangle}\right), \quad (10)$$

where $z_i$ represents an representation of an data instance $x_i$, which produces unit-norm, i.e. $\|z_i\|_2 = 1$. The $\langle z_i, z_a\rangle$ represents the positive pair and the negative pair $\langle z_i, z_b\rangle$ is the same. The method is to instantiate the weighting spherical density function constructed by the von Mises-Fisher distribution [14] with $\kappa > 0$. Inspired by the above, we aim to render the manifold representation located on the hypersphere $\mathcal{S}^d$. For a given pair of points $\mathcal{M}_1$ and $\mathcal{M}_2 \in \mathcal{S}^d$, we employ cosine similarity to compute the angle between manifold representations $\mathcal{M}_1$ and $\mathcal{M}_2$:

$$\text{dist}(\mathcal{M}_1, \mathcal{M}_2) = \cos\theta_{1,2}, \quad (11)$$

where $\theta_{1,2}$ is the angle between the manifold representation $\mathcal{M}_1$ and $\mathcal{M}_2$. Hence, the instance decision boundary of $\mathcal{M}_i^m$ for prior $a$ and biases $i$ is $\theta_{i+m[i],a} > \theta_{i+m[i],i}$, where $a$ and $i$ are indexes to positive and negative instances, respectively. This is essentially consistent with the prior that the data is distributed on the manifold:

$$L_{\text{BDC}} = \log \frac{e^{\left(\cos(\theta_{i+m[i],a})/\tau\right)}}{e^{\left(\cos(\theta_{i+m[i],a})/\tau\right)} + \sum_{j\neq i}^{n} e^{\left(\cos(\theta_{i+m[i],i})/\tau\right)}}. \quad (12)$$

In this loss function, the decision boundary of $\mathcal{M}_i^m$ concerning the biases and priors is defined by $\theta_{i+m[i],i}$ and $\theta_{i+m[i],a}$. This effectively drives $\mathcal{M}_i^m$ closer to the region corresponding to the correct class $\theta_{i+m[i],a}$. Therefore, this helps calibrate instance shifts caused by bias. In the ideal case, manifold instances are distributed on the unit sphere following instance semantics.

**Prior-Aware Debias Constraint (PDC):** As mentioned before, **BDC** effectively calibrates instance shift by leveraging bias-aware constraint in spherical space. We rethink our **BDC** method, which relies on prior conditions in training, potentially resulting in "Hallucination distribution" problems under distribution shift. In a biased training environment, the predicted answers will be similar to the distribution in the training set [18, 21, 33]. In fact, these manifold representations should be semantic correlations between instances and classes rather than fitting training distributions.

Therefore, We focus on mitigating prior dependence and optimizing the manifold representations by perceiving the distribution from the prior within the mini-batch. The relationship between the prior and manifold representation is reflected in the Kullback-Leibler divergence [54] $D(\cdot\|\cdot)$ of the probability distribution:

$$D\left(p_i^a \| p_i^m\right) = \sum_{i=1}^{N} P\left(s\left(\cos\theta_i^a\right)\right)\log \frac{P\left(s\left(\cos\theta_i^a\right)\right)}{P\left(s\cos\left(\theta_i + m[i]\right)\right)}, \quad (13)$$

where $p_i^a$ and $p_i^m$ are the probability distributions in a mini-batch, respectively. Previous studies reduce distribution dependence by

**Table 1: To verify the effectiveness of the combat hallucinations, experimental results on the VQA-CP v2 test set and VQA-CP v1 set with artificially change the prior and comparisons with state-of-the-art methods are presented.**

| Datasets | | VQA-CP v2 | | | | VQA-CP v1 | | | |
|---|---|---|---|---|---|---|---|---|---|
| Methods | | Overall-CP | Y/N-CP | Num-CP | Others-CP | Overall-CP | Y/N-CP | Num-CP | Others-CP |
| UpDn [2] | CVPR'18 | 39.74 | 42.27 | 11.93 | 46.05 | 37.96 | 42.79 | 12.41 | 42.53 |
| AdvReg [40] | NeurIPS'18 | 41.17 | 65.49 | 15.48 | 35.48 | 43.43 | 74.16 | 12.44 | 25.32 |
| RUBi [5] | NeurIPS'19 | 44.23 | 67.05 | 17.48 | 39.61 | 50.90 | 80.83 | 13.84 | 36.02 |
| LMH [11] | EMNLP'19 | 52.45 | 69.81 | 44.46 | 45.54 | 55.27 | 76.47 | 26.66 | 45.68 |
| GGE-iter [20] | ICCV'21 | 57.12 | 87.35 | 26.16 | 49.77 | 59.82 | 85.52 | 28.93 | 46.67 |
| AdaVQA [18] | IJCAI'21 | 54.67 | 72.47 | 53.81 | 45.58 | 61.20 | 91.17 | 41.34 | 39.38 |
| COB [25] | WACV'23 | 57.53 | 88.36 | 28.81 | 49.27 | 60.98 | 87.41 | 32.02 | 46.34 |
| PWVQA [45] | TMM'24 | 59.06 | 88.26 | 52.89 | 45.45 | - | - | - | - |
| GENB [10] | CVPR'23 | 59.15 | 88.03 | 40.05 | 49.25 | 62.74 | 86.18 | 43.85 | **47.03** |
| GGD [21] | TPAMI'23 | 59.37 | 88.23 | 38.11 | 49.82 | - | - | - | - |
| CVIV [38] | TMM'24 | 60.08 | 88.85 | 40.77 | 50.30 | - | - | - | - |
| RMLVQA [4] | CVPR'23 | 60.41 | **89.98** | 45.96 | 48.74 | 63.52 | **91.24** | 38.55 | 45.52 |
| MSCD | Ours | **62.29** | 88.28 | **55.45** | **50.54** | **65.60** | 90.49 | **50.51** | 46.91 |

manually adding annotations [7, 28, 31], which adds more additional costs. To make the predictions generated by the manifold representation statistically close to being independent of the prior, forcing it to focus on the semantics, using probability distribution estimation [26, 29, 54] is more general and more conducive to generalization. Therefore, we propose a prior-aware debias constraint to address this issue. The formulation between instance and prior is as follows in Spherical space:

$$\mathcal{L}_{\text{PDC}} = \frac{1}{2N} \sum_{i=1}^{2N} \left( D\left(\boldsymbol{p}_i^m \| \boldsymbol{p}_j^a\right) + D\left(\boldsymbol{p}_j^a \| \boldsymbol{p}_i^m\right) \right), \quad (14)$$

where $N$ is the number of prior and manifold representations in a mini-batch. In contrast, for an initialized spherical space, spherical constraints can force the model to learn semantic representations by reducing distribution dependence and bias effects.

## 3.4 Euclidean Debias Learning

Considering the synergistic properties of multi-space, for each instance, a homeomorphism can be established between Spherical space and Euclidean space. Since different spaces share joint representations, we can further combat the hallucination problem caused by bias through intrinsic connections. Inspired by [4, 32, 46, 49], we use overfitting characteristics to perform robust learning.

**Stage I: Bias-Examples Injection (BEI):** Following the previous setting [4], the bias-examples injection component is a classifier appended to the features $\mathcal{R}_i$ trained using the standard CE loss.

$$\mathcal{L}_{\text{BEI}} = - \sum_{\hat{\mathcal{A}} \sim |\mathcal{A}|} f(\hat{\mathcal{A}}) \log f(\hat{\mathcal{A}}), \quad (15)$$

where $\hat{\mathcal{A}}$ represents the biased prediction of the joint representation $\mathcal{R}_i$. The **BEI** combined with **BDC** can cluster instances in the feature space based on the source of bias to capture the complete semantic structure information while reducing instance offset.

**Stage II: Modal-Counterexamples Injection (MCI):** However, since bias comes from training data, too much **BEI** may exacerbate distribution dependence to cause adverse effects. To balance the pros and cons, inspired by [31, 50], we extend the Euclidean space to generate more modal counterexamples for training. Specifically, we construct vision counterexamples $(\mathcal{V}_i^-, \mathbf{Q}_i, \hat{\mathcal{A}})$ and questions

counterexamples $(\mathcal{V}_i, \mathbf{Q}_i^-, \hat{\mathcal{A}})$ for each instance by randomly generating and selecting in a mini-batch. Intuitively, in VQA, a question can only be answered if the question and image correspond. When provided with a counterexample instance as input, the VQA model fails to provide the correct answer by minimizing the loss.

$$\mathcal{L}_{\text{MCI}} = f\left(P\left(\mathcal{A} \mid \mathcal{V}_i^-, \mathbf{Q}_i\right)\right)[i] + f\left(P\left(\mathcal{A} \mid \mathcal{V}_i, \mathbf{Q}_i^-\right)\right)[i], \quad (16)$$

where $f(\cdot)$ represents the softmax function and $i$ is the index of ground-truth answer $\mathcal{A}_i$ in the answer set $|\mathcal{A}|$. Minimizing $L_{\text{MCI}}$ will encourage the prediction distribution to be far away from the true answer distribution of the bias due to the lack of supporting visual or question information. The **MCI** combined with **PDC** can encourage the model to focus more on the overall semantics of the instance, rather than based on the specific prior shortcut.

$$\mathcal{L}_{\text{EDL}} = \mathcal{L}_{BEI} + \text{W}(\text{epoch}) \cdot \mathcal{L}_{\text{MCI}}, \quad (17)$$

where $\mathcal{L}_{\text{EDL}}$ combines the CE loss of the training bias and modal counterexample injection components. The W(epoch) sets the period after the epoch from 0 to 1.

## 3.5 Ensemble Co-Bias Methods

Finally, we combine these components in a multi-space co-debias framework that can systematically work as a whole and mutually benefit from each other by integrating logits from the spaces to achieve robust representation learning and prior reduction. Therefore, the weighted sum of the four regularization losses and the basic VQA classification loss constitutes the total loss:

$$\mathcal{L}_{\text{Total}} = \mathcal{L}_{\text{SPH}-\text{M}} + \lambda_1 \mathcal{L}_{\text{BDC}} + \lambda_2 \mathcal{L}_{\text{PDC}} + \lambda_3 \mathcal{L}_{\text{EDL}}, \quad (18)$$

where $\lambda_1$, $\lambda_2$ and $\lambda_3$ are the hyperparameter settings of different components respectively.

## 4 EXPERIMENTS

### 4.1 Datasets and Evaluation Metric

The hallucination problem within the robust VQA task is challenging, and models that cleverly exploit biases or shortcuts may generate predictions that influence human decisions. Therefore, we select the VQA-CP v1 and VQA-CP v2 [1] datasets as benchmarks to evaluate the performance under changed prior conditions. All experiments adopt the standard evaluation metric [3].

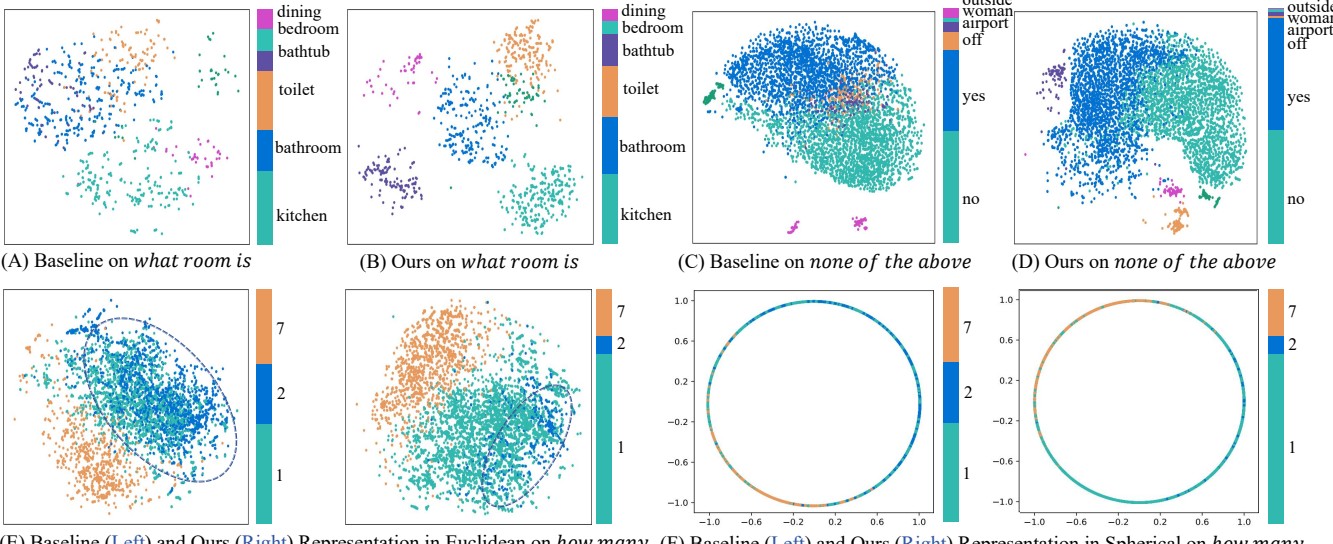

(A) Baseline on *what room is*  (B) Ours on *what room is*  (C) Baseline on *none of the above*  (D) Ours on *none of the above*

(E) Baseline (Left) and Ours (Right) Representation in Euclidean on *how many* (F) Baseline (Left) and Ours (Right) Representation in Spherical on *how many*

**Figure 5: In the figures (A)-(D) above, the answers embedded in the VQA cp v2 test set are compared with the baseline under different question types. The figure below is a visualization of the mitigation of the hallucination distribution.**

## 4.2 Baselines and Implementation Details

To verify the efficiency and generalization of our proposed model across two sets of biased VQA datasets, we select the most relevant works (Biases Mitigation Methods) for comparison. More baselines details will be introduced in the supplementary materials.

We implemented our MSCD framework in PyTorch with a single RTX 3090 GPU, and used the AdamW optimizer with weight decay 0.001. The learning rate is set to 0.001. The batch size $B$ is set to 512. The values of all hyperparameters including $\lambda_1$, $\lambda_2$, $\lambda_3$ in MSCD are set to 5.0, 1.0 and 5.0, respectively.

## 4.3 Experimental Results

As shown in Table 1, Our method achieves the highest overall accuracy, significantly outperforming all state-of-the-art baselines, across two VQA-CP benchmark datasets. These two datasets are specially designed to assess the ability to address the VQA hallucination problem, and the comparison results demonstrate the effectiveness of our method in effectively tackling these challenges. We observe that MSCD achieves gains of at least 1.88% and 2.25% respectively against the state-of-the-art methods. It is worth noting that the MSCD method achieves state-of-the-art performance on *Num* and *Other*, which have more demanding inference requirements. On VQA-CP v2, *Num* is significantly improved by 9.54%, which is exactly what this work wants to see. Due to the presence of hallucinations, simpler type problems such as *Yes/No* correlations are easily distinguished by bias. By simply and directly reducing bias, the performance of *Yes/No* may be improved, but it will also weaken the model's reasoning ability to a certain extent.

## 4.4 Ablation Study

For the multi-space co-debias method, the debias effect and cooperation of each component should be guaranteed in Table 2. To demonstrate the performance of each component in our MSCD

**Table 2: Ablation studies on the different settings of tasks.**

| Methods | AMM | BDC | PDC | EDL | Overall-CP |
|---|---|---|---|---|---|
| Baseline | - | - | - | - | 58.26 |
| w/ Spherical AMM | ✓ | - | - | - | 60.55 |
| w/ Spherical BDC | ✓ | ✓ | - | - | 61.08 |
| w/ Spherical PDC | ✓ | - | ✓ | - | 60.86 |
| w/ Sph-debias | ✓ | ✓ | ✓ | - | 61.70 |
| Co-debias (Ours) | ✓ | ✓ | ✓ | ✓ | **62.29** |

framework, 1) **Baseline**: This general UpDn architecture is used as a baseline model [2]. 2) **Baseline w/ Spherical AMM**: We train the frequency-instance margin model to initially construct the sphere space [4]. 3) **Baseline w/ Spherical BDC**: Based on the model, we built **BDC** and achieved an Overall-CP performance improvement of 0.53%. It reflects the superiority of **BDC** in alleviating instance offsets and effectively calibrate the offset instances to improve robustness. 4) **Baseline w/ Spherical PDC**: Achieved an Overall-CP performance improvement of 0.31%, showing an improvement in robustness against dependence on prior distributions, and the effect of motivating the model to pay attention to semantics is positive. 5) **Baseline w/ Sph-debias**: Combined with **BDC** and **PDC** constraints based on spherical space provides a more positive contribution to reduce instance offset and distribution offset to achieve better performance. 6) **Baseline w/ co-debias**: Based on the optimal performance of co-debias, we conclude that co-debias in different spaces can further improve the robustness to combat VQA hallucinations. Therefore, the above experimental results primarily prove the superiorities of the multi-space co-debias learning.

In order to further verify the superiority of our paradigm, we introduce bias branch [20] and impose **BDC** and **PDC** constraints on Euclidean embedding and Euclidean prior embedding. Following the same setting and architecture, the results all have different degrees of decline compared to multi-space co-debias learning.

**Table 3: Effectiveness of multi-space co-debias learning (The ∗ refers to BDC and PDC in Euclidean space) and Parameter Experiment of MCI epoch setting.**

| (A) | Methods | Overall | Yes/No | Num | Other |
|---|---|---|---|---|---|
| Euc-debias | BDC∗ | 60.19 | 89.14 | 45.77 | 49.59 |
| | (BDC+PDC)∗ | 60.69 | 89.96 | 48.83 | 47.96 |
| | (BDC+PDC+EDL)∗ | 60.59 | 88.92 | 46.95 | 49.48 |
| Co-debias | BDC | 61.08 | 89.84 | 47.17 | 49.81 |
| | (BDC+PDC) | 61.70 | 88.41 | 53.71 | 49.88 |
| | (BDC+PDC+EDL) | **62.29** | 88.28 | **55.45** | **50.54** |
| (B) | Epoch Setting | Overall | Yes/No | Num | Other |
| MCI | 3 | 61.90 | 88.63 | 54.89 | 49.82 |
| | 5 | 62.02 | 88.57 | 55.12 | 50.01 |
| | 8 | **62.29** | 88.28 | **55.45** | **50.54** |
| | 10 | 62.07 | 88.52 | 55.07 | 50.13 |
| | 15 | 61.96 | 88.51 | 55.00 | 49.96 |

**Table 4: Role of $\lambda_2$ in the training stage PDC of Sph-debias.**

| $\lambda_2$ | Overall-CP | Y/N-CP | Num-CP | Others-CP |
|---|---|---|---|---|
| -1 | 49.35 | 68.58 | 18.64 | 47.70 |
| -0.1 | 58.15 | 89.17 | 48.63 | 33.56 |
| 0.1 | 60.73 | 89.36 | 49.78 | 48.73 |
| 0.5 | 60.83 | 89.24 | 50.71 | 48.73 |
| 1 | **61.70** | 88.41 | **53.71** | **49.88** |

## 4.5 Parameter Sensitivity

We verify the two most important hyperparameters in the model, The first one is about the $\lambda_2$ setting in **PDC**, which represents the correlation between the learned manifold embedding and the prior distribution. As shown in Table 4, the unbalanced learning process will bias the learned feature representation towards biased data, and when $\lambda_2$ in PDC is a negative number, the model performance will drop sharply. This is mainly because when $\lambda_2$ is negative, the distribution represented by the manifold depends more on the prior. When $\lambda_2$ is positive, especially when it is 1, the model explicitly learns semantics that focuses on the correlation between instances and classes. The second one is about the setting of *Epoch* in **MCI**. This parameter controls the timing of modal counterexample injection. As illustrated in Table 3 (B), we quantitatively study the impact of *Epoch* epochs of **MCI**. This phenomenon can be attributed to the fact that after Spherical debias learning, the high-confidence model can improve the model's sensitivity to semantics by further learning error semantics, thereby reducing its reliance on training priors. When *Epoch* exceeds 8, performance will decrease slightly. A too late priming period will inevitably weaken the positive impact of differential learning and thus produce suboptimal results.

## 4.6 Visualization Results

**Featured Space Visualization:** As shown in Figure 5, we visualize the feature distribution of each instance for two question types [6]. It can be noticed that the instances learned by our model in different question types are more closely related to the answer class to which they belong, making it simpler to distinguish the answer class corresponding to each instance. This reduces instance shift and mitigates VQA hallucinations. In addition, we focus on visualizing the answer distribution of question types in Figure 6. The training set and test set distributions for these types are noticeably

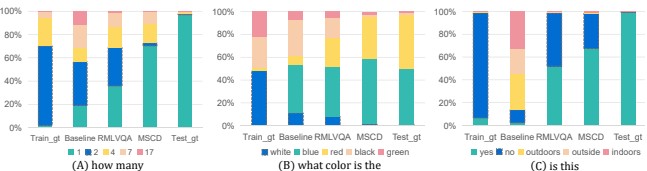

(A) how many    (B) what color is the    (C) is this

**Figure 6: Illustration of combating VQA hallucinations. We visualized the distribution of answers. the MSCD shows consistent improvements to Baseline and RMLVQA [4], effectively alleviating the hallucinations distribution problem.**

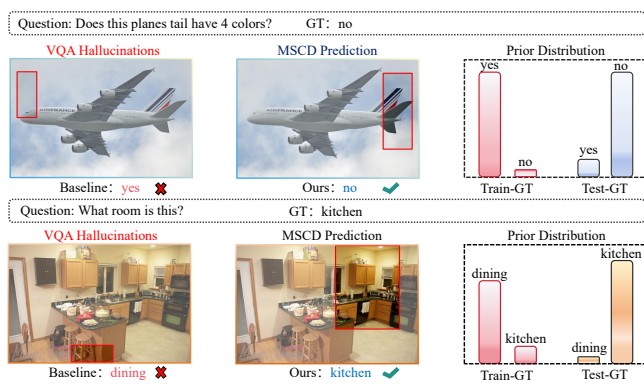

**Figure 7: Visualization analysis of our method. Two examples were chosen to show the robustness of the MSCD.**

different. Our method effectively mitigates VQA hallucinations, which is the focus of this article. In contrast, our MSCD method achieves satisfactory hallucination mitigation effects through the multi-space co-debiasing learning paradigm.

**Attention Region Visualization:** From the result analysis in Figure 7, due to the influence of biases, baseline failed to find the target object mentioned in the problem in the image, resulting in erroneous predictions. Taking the "plane" picture as an example, the model will naturally generate an illusive answer to "yes" instead of paying attention to the overall semantics of the question and answer. In contrast, our MSCD demonstrates the ability to mitigate hallucinations and enable the modality to focus on semantics to accurately locate the target object with a high degree of confidence, thereby giving precise answers to the questions posed.

## 5 CONCLUSION

This paper establishes a novel robust visual question answering paradigm (MSCD), which simultaneously utilizes multi-space co-debiasing in a unified framework to combat VQA hallucinations. We construct bias-aware and prior-aware constraints in spherical space to improve robustness by alleviating bias effects and distribution dependence, encouraging the model to focus on the semantics of instances. Switching to the space homeomorphism perspective, we employ bias example and modal counterexample learning strategies through the over-fitting characteristics of Euclidean space to further assist robust learning. Experiments conducted on three widely used datasets demonstrate the encouraging performance of our approach compared to existing state-of-the-art methods.

## ACKNOWLEDGMENTS

This work was supported in part by the National Natural Science Foundation of China (Grant Nos. 62302172, 62176077), in part by the Guangdong International Science and Technology Cooperation Project (Grant No. 2023A0505050108), in part by the Shenzhen Key Technical Project (Grant Nos. JSGG20220831092805009, JSGG20201103153802006), and in part by the Opening Project of Guangdong Province Key Laboratory of Information Security Technology (Grant No. 2023B1212060026).

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
