# OpenReview forum: "Combating Visual Question Answering Hallucinations via Robust Multi-Space Co-Debias Learning"
_acmmm.org/ACMMM/2024/Conference — MM2024 Poster_

### Official Review · Reviewer_QiL5 · 2024-05-20

**Rating:** 3
**Confidence:** 3

**Summary:**

This paper introduces MSCD, a novel robust visual question answering paradigm that combats VQA hallucinations. By utilizing multi-space co-debias and constructing bias-aware and prior-aware constraints in Spherical space, MSCD improves robustness by mitigating bias effects and distribution dependencies. Leveraging Euclidean space's overfitting characteristics, the model employs bias example and modal counterexample learning strategies for further robust learning. Experimental results on three datasets demonstrate MSCD's effectiveness, outperforming existing state-of-the-art methods in VQA.

**Strengths:**

1.	This paper presents MSCD, a novel approach for combating VQA hallucination by integrating Spherical and Euclidean spaces into a unified framework. The method introduces innovative bias-aware and prior-aware debias constraints, specifically designed for spherical debias learning, to address instance shift and distribution shift, effectively alleviating VQA hallucinations. Furthermore, MSCD's two-stage strategy in Euclidean space enhances spherical debias learning by uncovering prior correlations and modality-semantics interplay.
2.	Extensive experiments on benchmark datasets demonstrate MSCD's effectiveness, achieving state-of-the-art performance and showcasing its potential in addressing VQA hallucination issues stemming from biases.

**Limitations:**

1.	The paper is confusing in certain areas of writing. For instance, there is no clear explanation for the letter "B" in Equation 3, or for "b_i" and "b_j" in Equation 5, among others. Additionally, there are several typos, such as "night" being written as "nigh" in line 107. These issues in writing quality somewhat affect the readability and comprehensibility of the paper, especially due to the lack of explanations for specific symbols in the equations.
2.	There seem to be some errors in the experimental results section. For example, in Table 1, the performance of AdaVQA in the Y/N-CP column for VQA-CP v1 (91.17) is better than MSCD (90.49) as presented in the paper, but the results for MSCD are incorrectly bolded. Similarly, in Table 2, the performance of UpDn and LMH in the Others column (55.66 and 54.69) is better than GGD (54.66), yet the results for GGD are bolded. These errors or confusing points detract from the quality of the paper and need to be corrected or further explained.
3.	Table 2 shows that MSCD's performance on the Overall metric in VQA v2 is lower than many baseline methods, although it performs better on VQA-CP v2. Does this indicate that the MSCD method sacrifices performance on standard VQA v2 to improve performance on OOD benchmarks? In other words, is this method a trade-off approach designed to enhance the model's performance on OOD benchmarks?

**Suitability:**

3

---

### Official Review · Reviewer_EnLz · 2024-05-20

**Rating:** 4
**Confidence:** 3

**Summary:**

This paper addresses the challenge of bias in visual question answering (VQA) which can cause semantic ambiguities and generate shifts in the feature space of VQA instances leading to VQA Hallucinations. To combat this issue, a Multi-Space Co-debias Learning (MSCD) approach is proposed that effectively mitigates bias-induced instance and distribution shifts in multi-space under a unified paradigm. The approach uses bias-aware and prior-aware debias constraints utilizing the angle and angle margin of the spherical space to construct bias-prior-instance constraints, further enhancing the robustness of representations. Additionally, inherent overfitting characteristics of Euclidean space are leveraged to introduce bias components from biased examples and modal counterexample injection, further assisting in multi-space robust learning. The proposed method generates more robust representations that effectively mitigate biases to combat hallucinations. Experiments conducted on multiple benchmark datasets demonstrate that the proposed MSCD method outperforms state-of-the-art baselines.

**Strengths:**

1. The paper is well written.

2. The idea of multi-space co-debias learning sounds interesting.

3. The experiment results well demonstrate the effectiveness of the proposed method.

**Limitations:**

1. Figure 1 is not well understood. Does it imply that the model exhibits inconsistent levels of hallucination between the training and test sets?

2. It is recommended that the paper includes a sensitivity analysis for lambda_1 and lambda_3 to better understand their impact on the model's performance.

3. Are methods like VCD, which are designed to address multimodal large models, applicable to this problem? If so, what are the effects or improvements they bring to the issue?

**Suitability:**

3

---

### Official Review · Reviewer_xbLx · 2024-05-21

**Rating:** 2
**Confidence:** 4

**Summary:**

This paper tackles debiasing of VQA models through a multi-space co-debias approach.
It takes the issue of VQA models being biased to in-distribution samples and tries to tackle hallucination through different losses that aid in reducing the biases.

**Strengths:**

The performance of the work in the specified field and given experiments are quite high.
The idea they present is quite simple yet effective.

**Limitations:**

There are a few concerns to this work.
First of all, which is the baseline that was used in this paper?
It does not say what the baseline is and it seems that the method that the authors are proposing only seems to work on whatever their baseline is. Although in the Ablation section, they claim to integrate the model [4], the overall CP score does not align with the main table.
Because we do not know what the architecture is based of off, it is hard to say this is a fair comparison.

Also, the method seems to be built off of added loses on other methods, but how does the method work on other architectures or baselines?
This would be very helpful in understanding the contribution of this work.

How about other architectures like the LXMERT baseline?
With UpDn being such an old model now, there should be some comparisons to a Transformer based model.

There also has been a lot of controversy regarding the use of VQA-CP2 as a method of testing biases in VQA models, and this paper
On the Value of Out-of-Distribution Testing: An Example of Goodhart's Law, has advocating the use of other baselines to test true ID-OOD settings.
Are there any scores available for GQA-OOD, VQA-CE, or such?

Also there are a lot of ?s in the figures.

**Suitability:**

3

---

### Official Review · Reviewer_nQhM · 2024-05-27

**Rating:** 5
**Confidence:** 2

**Summary:**

This paper proposes a robust Multi-Space Co-debias Learning (MSCD) approach for combating VQA hallucinations. This method addresses the problem by integrating spherical and Euclidean spaces into a unified co-debiasing framework. It deploys a two-stage strategy where the Euclidean space assists spherical debiasing to expose prior correlations and modality-semantics interplay. Experiments conducted on two biased benchmark datasets and a balanced dataset demonstrate the effectiveness of the proposed MSCD method in combating VQA hallucinations.

**Strengths:**

1.	The paper presents a novel multi-space co-debias learning to address the issue of VQA hallucination stemming from biases.
2.	The experiments are solid. The model shows clear improvements over existing models on the out-of-distribution benchmark datasets.
3.	The writing is easy to follow. The arguments are coherent and logically structured. The figures and tables are well-designed and effectively illustrate the points.

**Limitations:**

1. As shown in Table 2, the proposed method performs well on the out-of-distribution dataset but is weaker than many existing methods on the standard VQA dataset. Please provide a detailed explanation and analysis.
2. The visualization examples are insufficient. In Figure 7, please provide examples for each type of question, especially for the "how many" questions mentioned in the introduction.

**Suitability:**

3

---

### Meta-Review · Area_Chair_AKd3 · 2024-07-02

**Recommendation:** Accept (Poster)
**Confidence:** 4

**Metareview:**

The initial ratings are 1 WR, 1 BR, 1 BA and 1 WA, and they are 3 BA and 1 WA after rebuttal. The reviewers all appreciated excellent writing, clear logic, substantial content, detailed exposition, and are satisfied with the authors' responses. The reviewers all appreciated excellent writing, clear logic, substantial content, detailed exposition, and are satisfied with the authors' responses.